
**The impact of long-term changes in ocean waves and storm surge on coastal shoreline**
**change: A case study of Bass Strait and south-east Australia**
Mandana Ghanavati[1], Ian R. Young*[1], Ebru Kirezci[1], Jin Liu[1]
[1]Department of Infrastructure Engineering, University of Melbourne, Melbourne, VIC 3010, Australia.
* Corresponding author: ian.young@unimelb.edu.au
**Abstract**
Numerous studies have demonstrated that significant global changes in wave and storm surge
conditions have occurred over recent decades. Climate projections indicate such changes are likely
to continue out to at least 2100. As coastlines respond to the environmental forcing of waves and
storm surges, the question of whether the observed and projected changes in waves and storm
surges, will impact coastlines in the future, is important. Previous global-scale analyses of these
issues have been inconclusive. This study investigates the south-east coast of Australia over a
period of 26 years (1988-2013). Over this period, this area has experienced some of the largest
changes in wave climate of any coastal region, globally. The analysis uses high-resolution hindcast
data of waves and storm surge, together with satellite observations of shoreline change. All
datasets have been previously extensively validated against in situ measurements. The results show
that beaches along this region appear to have responded to the increases in wave energy flux and
changes in wave direction. This has enhanced non-equilibrium longshore drift and recession of the
coastline, with recession rates of up to 1m/year.
**1. Introduction**
Sandy coastlines are dynamic systems, responding to changes in waves, storm surge, sea level,
available coastal sediment supply and human activities (e.g. coastal structures, beach nourishment)
(Komar, 1998; Masselink, et al., 2016). These changes occur on a variety of spatial and temporal
scales. Spatially, changes in beach alignment and the presence of coastal shoreline features
(headlands and bays) impact both the wave climate for individual beaches and the characteristics
of longshore drift. At temporal scales of days, beach erosion results from individual storms
(Komar, 1998; Harley, et al., 2017; Masselink, et al., 2016). At time scales of 2 to 10 years,
changes in storminess associated with climate indices (e.g. El Niño) (Ranasinghe, et al., 2004;
Harley, et al., 2011; Barnard, et al., 2015; Vos, et al., 2023) impact beach systems. Longer term
changes in mean sea level as a result of climate change are also predicted to result in coastal
recession (Hinkel, et al., 2013; Ranasinghe, 2016; Vousdoukas, et al., 2020; Ranasinghe, et al.,
2021; Vitousek, et al., 2023). It should be noted that throughout this paper we refer to shorter-term
changes in beach location due to storms or a series of storms as erosion or accretion. Longer-term
changes such as those due to climate change are referred to as recession or progradation.
Waves and storm surges are generated by environmental variables (wind and sea level pressure
gradient). It has been shown that these environmental variables are impacted by climate change





and hence long-term historical changes (trends) in waves (Wang & Swail, 2001; Wang, et al.,
2009; Hemer, 2010; Young, et al., 2011; Aydoğan & Ayat, 2018; Zheng & Li, 2017; Young &
Ribal, 2019; Takbash & Young, 2020; Reguero, et al., 2019; Cao, et al., 2021) (Young & Ribal,
2022; Liu, et al., 2022; Morim, et al., 2022; Erikson, et al., 2022) and storm surges (Paprotny,
2014; Androulidakis, et al., 2015; Cid, et al., 2016; Muis, et al., 2016; Kim, et al., 2017; Feng, et
al., 2018; Ghanavati, et al., 2023) have been observed. A number of studies have also projected
continued global increases (positive trends) in wave height over the 21$^{st}$ century, particularly in
the Southern Hemisphere, under plausible climate change scenarios (Hemer, et al., 2013; Meucci,
et al., 2020; Hochet, et al., 2021; Liu, et al., 2022; Meucci, et al., 2023; Morim, et al., 2023; Liu,
et al., 2023).
If sandy coasts are impacted by changes in wave and storm surge conditions, the potential for
continued increases in the values of these variables in the future, raises the question as to what
impact this may have on sandy coastlines and associated communities. As a means of determining
potential future impacts, the obvious precursor is to assess the impacts that historical changes in
long-term wave and storm surge conditions have had on coastlines. Ghanavati, et al. (2023) have
investigated this issue at global scale by using long-term modelled wave and storm surge data
together with satellite observations of beach recession/pogradation over the last 30 years. They
found that, noting the relatively small trends in wave and storm surge conditions over this period,
the accuracy of the available data, and other unrealated impacts on shoreline response (e.g.
availability of sediment, human impacts), no clear relationship was evident.
The present study extends the Ghanavati, et al (2023) work by examining the south-east coastline
of Australia in detail. This is an area where long-term trends in wave conditions are some of the
largest in the world, responding to changes in wave climate in the Southern Ocean (Liu, et al.,
2022). As a regional area is considered, it is possible to use higher resolution data (both model and
satellite) removing uncertainties in the global-scale Ghanavati, et al (2023) study.
The structure of the paper is as follows. Section 2 outlines the study area, data sets and analysis
techniques used in the study. Results are given in Section 3, including the observed relationships
between changes in wave and storm surge quantities and beach recession/progradation. Discussion
and conclusions are provided in Section 4.

## 2. Methodology


### 2.1 Study Area

The study region is shown in Figures 1 and 2, and covers an area of 137E°–155°E, 35S°–45°S.
Three Australian coastal states span this domain, Victoria, southern New South Wales and the
island of Tasmania in the south of the domain. The south-eastern coast of the mainland of Australia
(Victoria), the coastal area of the study, is separated from Tasmania by the relatively shallow Bass
Strait. The area is exposed to a particularly complex wave climate (Liu, et al., 2022). To the west,


the coast is exposed to the Southern Ocean and hence experiences a very energetic wave climate
with recorded significant wave height as high as 10m (Meucci, et al., 2023). The wave climate of
this region is dominated by south-westerly Southern Ocean swell. Central regions of the study
domain are protected by the island of Tasmania and have a mixed wave climate with swell from
both the south-west and south-east and locally generated wind sea. To the east, the wave climate
is more heavily dependent on the local wind-sea but with south-easterly swell still playing a role
(Liu, et al., 2022).
Both observational data from satellite altimeters (Young, et al., 2011; Young & Ribal, 2019;
Timmermans, et al., 2020) and model hindcasts (and reanalyzes) (Cao, et al., 2021; Young &
Ribal, 2022) show that over the last 35 years, there has been a small global increase in mean
significant wave height. This increase is largest in the Southern Ocean (approximately 3mm/year
or an increase of 3% over the last 30 years), which results in impacts across the Indian, South
Pacific and South Atlantic Oceans due to radiating swell. Therefore, the study area is a location
where relatively large changes in significant wave height have occurred over the period.

## *2.2  Datasets*

This study uses regional datasets for each of wave, storm surge, and coastal change from which
the historical trend magnitudes of the various quantities were calculated. The datasets under
consideration cover different periods of time, and thus, to ensure consistency across analyses, a
common time period from 1988 to 2013 was selected. A description of each dataset used in the
study is provided below.
*Liu et al. (2022) regional wave hindcast* is a high-resolution regional wave hindcast dataset based
on a WAVEWATCH III model with an ST6 physics package (Liu, et al., 2021). The regional
model covers the domain shown in Figure 2 using an unstructured grid with a coastal resolution
as small as 500m and a coarser deep water resolution as large as 10km. The regional model is
nested within a global model using the same ST6 physics (Liu, et al., 2021). Both the regional and
global models are forced with ERA5 winds (Hersbach, et al., 2020). The regional wave model
dataset has been extensively validated (Liu, et al., 2022; Liu, et al., 2023) against both a network
of coastal buoys and satellite altimeter data. Wave data were available from the hindcast with a
temporal resolution of 1 hour. The period of the hindcast was from 1981 to 2020. The dataset's
high resolution is particularly important for studying coastal regions, where wave conditions can
vary significantly over short distances. Additionally, the long period of coverage allows us to
identify and analyze trends in the wave climate over several decades, providing insight into the
possible effects of historical climate change on the region.
*Colberg, et al. (2018) Australian water level hindcast* is a dataset of sea level simulations for the
Australian coastline. The dataset was generated using the Regional Ocean Modelling System


(ROMS) (Shchepetkin & McWilliams, 2005), which was run in a depth-integrated form on a 5 km
resolution grid for the Australian region. Tidal boundary conditions were provided by the global
model TPXO7.2 (Egbert & Erofeeva, 2002). The ROMS model was run for the period 1981-2013
and was forced with NCEP Climate Forecast System Reanalysis (CFSR) (Saha, et al., 2010) wind
and surface pressure data. The model has been validated at 14 tide gauge locations around the
Australian coastline (Colberg, et al., 2018). Again, the output was available on an hourly basis.
*Bishop-Taylor et al., (2021) Geoscience Australia beach dataset* is a high-resolution regional
dataset of shoreline change rate for the coast of Australia. The dataset utilizes a combination of
satellite visual data and tidal modelling to map shoreline change, with an along-coast resolution of
30m for non-rocky (sandy or muddy) areas. The dataset provides annual values of the shoreline
position over the period 1988 to 2019. The dataset has been extensively validated using in-situ
measurements, comprising 330 validation transects, each spanning over 10 years of coastal
monitoring data (Bishop-Taylor, et al., 2021).

### 128    *2.3    Trend calculation*

Each of the datasets (waves, storm surge, shoreline location) are defined at different resolution and
in different manners (structured and unstructured grids, specific shoreline positions), therefore
none of these quantities are co-located. As shown by Ghanavati, et al., (2023) and subsequently
confirmed in Figures 3, 4 and 5, trends in both wave height and storm surge quantities generally
vary smoothly along extended coastal regions (100s of kilometres). Shoreline
recession/progradation rate can, however, vary rapidly in magnitude and sign over relatively short
spatial scales (10s of kilometres) (Luijendijk, et al., 2018; Ghanavati, et al., 2023). That is, one
beach can be receding whilst the next is prograding. As such, simple scatter plots of rates of change
of wave and storm surge quantities verses recession/pogradation rates are not meaningful. Rather,
one needs to consider relationships over spatial regions of the coastline. To achieve such an
analysis, we divide the study domain in Figure 1 into six regions, each spanning 2° in longitude –
(a) 138E°-140E°, (b) 140E°-142E°, (c) 142E°-144E°, (d) 144E°-146E°, (e) 146E°-148E° and (f)
148E°-150E° from west to east. These regions span the differing wave climates of the study
domain (see Figure 2 and subsequent discussion). For analysis purposes, we present data as
follows. Wave quantities are presented both as colour shaded plots, and at shoreline locations
corresponding to ocean points defined by the unstructured WAVEWATCH III computational grid.
Storm surge quantities are shown at the locations corresponding to the ocean points nearest the
land/sea transition of the ROMS 5km computational grid. Coastal change points are as defined at
coastal locations in the Bishop-Taylor et al., (2021) dataset, which has an along-cost resolution of
30m.
Each of the three datasets used in the study covers a different period of time: wave hindcast - 1981
to 2020, storm surge data - 1981 to 2013, and shoreline change data - 1988 to 2019. To ensure a
consistent evaluation of the trends and variability in the oceanic parameters, a common analysis
period of 1988 to 2013 was selected for the study.


For each of the datasets, a range of quantities to be investigated were calculated. These include:
waves – mean significant wave height ($H_s$), 95th percentile significant wave height ($H_s^{95}$), mean
wave energy flux ($C_g E$), mean wave period ($T_m$) and mean wave direction ($\theta_m$), where $C_g$ is the
group velocity of waves and $E = H_s^2 / 16$ is the wave energy. The hourly data from the regional
wave model was used to calculate annual values of each of these quantities.
As noted above, various datasets have different temporal and spatial resolutions and hence slightly
different approaches were used to evaluate the variability and extremes of oceanic parameters. The
wave and surge time series were collected at a temporal resolution of 1 hour, while the shoreline
dataset provided annual shoreline change with reference to the shoreline location in 2019.
Therefore, annual mean values of wave parameters including significant wave height, wave energy
flux, wave direction and wave period were calculated. Furthermore, the extremes were determined
by calculating annual higher percentiles (95th, 98th, and 99th) for significant wave height and
surge level. These metrics provide a consistent basis for evaluating the variability and extremes of
the oceanic parameters across different datasets. As the various percentile thresholds gave similar
results, extreme events were determined as occasions on which the time series exceeded the 95th
percentile but with such events separated by a minimum of 48 hours. The number of such events
in each year were defined as $N_{H_s^{95}}$. In a similar fashion, storm surges were defined as occasions
when the water surface elevation, $\eta$, exceeded the 95th percentile ($\eta^{95}$) and the number of such
events was defined as $N_{\eta^{95}}$. Again, annual values of these quantities were determined. The annual
values of shoreline position from the Bishop-Taylor et al. (2021) data were defined in a similar
manner and represented as $C_{GA}$.
The annual values of each quantity were then used to determine linear trends over the period 1988-
2013. Both linear regression and the non-parametric Tiel-Sen estimator (Sen, 1968) were used for
this purpose. As the resulting values were very similar, the Sen slope estimates are used in the
subsequent analysis. The resulting trend values are represented as: $\Delta H_s$, $\Delta H_s^{95}$, $\Delta C_g E$, $\Delta \theta_m$,
$\Delta N_{H_s^{95}}$; $\Delta \eta^{95}$, $\Delta N_{\eta^{95}}$; $\Delta C_{GA}$.
## 3. Results
### 3.1 Wave climate
Figure 2 shows the mean wave climate of the study area and how it has changed over the period
1988 to 2013 as indicated by the Liu, et al. (2022) hindcast. Figures 2a and 2b show the mean
significant wave height $\bar{H}_s$ and wave energy flux, $\overline{C_g E} = \rho g^2 H_s^2 T_m / (64\pi)$, respectively. As
noted above, the significant wave height and wave energy flux vary significantly across the study
area. In the west, the coastline is exposed to energetic Southern Ocean swell with mean $H_s$ of
approximately 3m. In the eastern regions of the study area, where there is protection provided by




the island of Tasmania, mean $H_s$ decreases significantly to less than 1.5m, a decrease by a factor
of approximately 2. The wave energy flux shows an even more significant change, with mean
values varying from approximately 60kW/m in the west to 15kW/m in the east, a factor of 4. The
substantial reduction in wave energy flux is attributed to the protection provided by the island of
Tasmania, which leads to a decrease in both $H_s$ and $T_m$. As shown by Liu, et al. (2022), the
mean/peak wave direction also changes significantly across the domain. In the west, the dominat
wave direction is defined by energetic south-westerly swell. In the east, the protection provided by
the island of Tasmania means that swell entering the area is predominately from the south-east.
The changes in wave climate over the study period are also significant across this region. As noted
above, a range of studies have shown that the Southern Ocean wave climate has increased over the
past 35 years (Young, et al., 2011; Young & Ribal, 2019; Cao, et al., 2021; Young & Ribal, 2022).
Swell from the Southern Ocean dominates the western areas of the study region and hence there
have been significant changes in the wave climate, as shown by Figures 2c-h. In the west, $H_s$ has
increased by approximately 5% (Figure 2c) over the study period and $C_g E$ by approximately 14%
(Figure 2d). In contrast, in the east, where the wave climate is not as exposed to Southern Ocean
swell, these values decrease to approximately zero (no change). Figures 2e and f clearly show that
the positive trends in $H_s$ are due to changes in both swell and local wind-waves. Figure 2g also
shows that there have been only small changes in $T_m$ across the domain.
The most dramatic changes in wave climate concern the mean wave direction, $\theta_m$. Over the
western regions of the study domain, there has been a small counter-clockwise rotation of the mean
wave direction (less than 1.5°). This is a result of the gradual southward movement of Southern
Ocean low pressure systems over recent decades (Morim, et al., 2022). This small change in deep
water wave direction, significantly impacts the shadow region in the lee of Tasmania and hence
the wave direction, resulting in much larger counter-clockwise rotations of approximately 5°
(Figure 2h). These values reduce towards the coast of mainland Australia (eastern area of study
region) but are still larger than 3°.
*3.2    Storm Surge Climate*
As noted above, storm surges were defined as events where the water surface elevation exceeded
the 95[th] percentile value, $\eta^{95}$. Figure 3 and 4 show plots for each of the sub-regions referenced in
Figure 1. These figures show colour contoured values of $\Delta C_g E$ (Figure 3) and $\Delta \theta_m$ (Figure 4),
coastal values of $\Delta \eta^{95}$ and $\Delta C_{GA}$. In contrast to the wave climate, changes in storm surge, $\Delta \eta^{95}$
are very consistent along the coastline of the study area. Values of $\Delta \eta^{95}$ are negative along the
entire coastline, decreasing in magnitude from approximately -0.3cm/year in the west to
-0.2cm/year in the east. The fact that the magnitude of storm surges has been decreasing over this
period is consistent with the observations of Liu, et al. (2023) that as Southern Ocean low pressure



systems move south, they increase the mean atmospheric pressure and reduce the pressure gradient
over southern Australia. As surface pressure (and wind) drives storm surge, this results in a
tendancy for a reduction in the magnitude of storm surges.

### *3.3    Relationship between waves, storm surge and shoreline change*
As previously shown at global scale by Luijendijk, et al. (2018) and Ghanavati, et al. (2023),
recession/progradation rates vary in magnitude and sign on relatively small spatial scales. This is
because sediment transport can be both offshore/onshore as well as longshore. In the case of non-
equilibrium longshore transport of sediment, one would expect some beaches to recede whilst
other receive sediment from these beaches and hence prograde. Ghanavati, et al. (2023) speculated
that coastlines which show such non-equilibrium behaviour may be responding to long-term
changes in the environmental forcing provided by trends in waves and storm surge. A causal
relationship is, however, complicated by other variables which may have a larger impact on beach
position. These additional factors include the availability of sediment supplied to beach
compartments from fluvial sources and the impacts of human-induced interventions such as coastal
structures and beach nourishment (Ranasinghe, 2016). Ghanavati, et al. (2023) limited
recession/progradation data to values in the range ±1m/year to confine the datasets to changes
which may be a result of long-term processes rather than fluvial and human-induced influences,
which tend to be much larger in magnitude (Luijendijk, et al., 2018).
Therefore, following these precidents, in Figures 3 – 6, the quantity $\Delta C_{GA}$ has been filtered to retain
only values in the range ±1m/year. Figure 5 shows values of $\Delta C_{GA}$ (in the range ±1m/year) as a
bar chart along the coastline from 138E° to 150E°. Each of the 2° regions shown in Figures 1, 4
and 5 is marked along the longitude axis. As expected, values of $\Delta C_{GA}$ in Figures 3, 4 and 5 show
both positive (progradation) and negative (recession) values. To quantify recession/progradation,
values of $\Delta C_{GA}$ in the range -0.05m/year to -1.00m/year are clasified as recession, values in the
range +0.05m/year to +1.00m/year as progradation and values in the range ± 0.05m/year as
representing stable coastlines. Table 1 shows the percentage of coastal locations classified as
receding, prograding or stable under these criteria. In addition, Figure 6 shows histograms of the
distribution of the magnitudes of the values of $\Delta C_{GA}$.
Table 1 and Figure 6 show that the sections (c) 142E°-144E° and (f) 148E°-150E° are
predominately receding. Segment (d) 144E°-146E° shows quite large values of both recession and
progradation (see Figure 5) but with more locations prograding than receding. However, this
region is complicated by the presence of Port Phillip Bay. The other segments (a), (b) and (e)
show no clear difference between the percentage of receding and prograding locations.
To understand the results shown in Table 1, we consider each of the two degree sections shown in
Figures 3, 4 and 5. In these figures, values of the trend in wave energy flux, $\Delta C_g E$ (Figure 3) or


wave direction, $\Delta\theta_m$ (Figure 4) are shown as colour shaded contours over the regions. The trend
in storm surge (always negative) are shown as colour coded squares at 5km intervals along the
shoreline, at the resolution of the water level model. The satellite-derived values of trend in
shoreline location at each beach location (Bishop-Taylor, et al., 2021) are shown as colour coded
filled circles, at the 30m along-coast resolution.
Figures 3a and 4a show the region from 138E° to 140E° (segment (a), Victor Harbour to Cape
Jaffa). This region shows relatively small positive values of $\Delta C_g E$ (approximately 0.01kWm⁻
¹/year) and a small counter-clockwise rotation of the mean wave direct (approximately
-0.02deg/year or 0.6° over 30 years). In response to these small changes in wave properties there
is no consistent changes in shoreline. In the western regions (138.6E°-139.2E°) the shoreline is
prograding. However, this may be associated with fluvial sediments, as this region is the ocean
entrance of Lake Alexandrina and the mouth of the Murray River. These results are consistent with
the bar chart of Figure 5 and the results in Table 1 and Figure 6a that there is no clear difference
between recession and progradation for segment (a).
Moving east to segment (b), values of $\Delta C_g E$ increase (Figure 3b) and the region shows small
receding shorelines (139.6E°- 141.0E°). This changes to progradation between 141.0E°-141.2E°,
west of Cape Bridgewater. This behaviour is consistent with sediment being moved along the
shoreline west to east from 139.6E°- 141E° by the increasing wave energy flux and the prevailing
wave direction from the south-west. This sediment transport is interrupted by Cape Bridgewater
resulting in the progradation between 140.8E°-141.2E°. The overall balance between these regions
results in no clear difference between locations receding and prograding in Table 1 and Figure 6b.
The strong positive trend in wave energy flux is maintained east of Cape Bridgewater (segment
(c), Figures 3c) with small counter-clockwise rotation of the mean wave direction (Figure 4c).
Along this extended region of the coast to Cape Otway (141.6E°-143.6E°), the coastline shows
small recession (approximately -0.1m/year – 3m over the measurement period of 30 years). East
of Cape Otway, the magnitude of the recession decreases and the shoreline shows little net change
in location. This behaviour is consistent with the reduced impact of south-westerly swell east of
Cape Otway, which provides some shelter from such waves. Table 1 and Figure 6c show that
summed across the full segment (c), a total of 53% of locations are receding and only 27%
prograding.
East of Cape Otway, the wave energy flux climate near the coast decreases (Figure 2b), as Cape
Otway provides protection from the south-westerly swell and $\Delta C_g E$ also decreases as the
protection provided by Tasmania becomes important (Figure 3d). The shoreline trends, $\Delta C_{GA}$, are
complicated by the presence of Port Phillip Bay (Figures 3d, 4d). From Cape Otway to Inverloch
(143.6E°- 145.8E°) there is relatively little change in $\Delta C_{GA}$. The relatively small region from
Inverloch to Wilson's Promontory (145.8E° - 146.4E°) shows a receding shoreline, previously
noted in studies of the area (Leach, et al., 2023). As a result, there is no clear overall differences


between recession and progradation for this section (Table 1 and Figure 6d). However, if one
considers just the ocean beaches (exclude Port Phillip Bay in Figures 3d and 4d), then there is
small recession along the entire coastline of section (d).
East of Wilson's Promontory the coastline is characterized by very long beaches and barrier islands
(Ninety-mile beach). This region from 147E° to 149.6E° (Wilson's Promontory to Cape Howe)
(Figures 3e-f, 4e-f) is characterized by a large counter-clockwise rotation of the mean wave
direction. The region immediately east of Wilson's Promontory (146.5°E – 147°E) shows strong
progradation. The remainder of this extended coastline, however, shows consistent recession of
approximately -0.5m/year (15m over the measurement period), particularly for section (f). This
section shows the strongest recession of any extended section, with Table 1 showing 60% of
locations receding and only 30% prograding. As noted above, the dominant swell in this region is
from the south-east and, although the changes in wave energy flux are small, there has been a
significant counter-clockwise rotation of the wave direction over the study period. This results in
the wave direction gradually becoming more shore-parallel. Therefore, the shoreline change noted
above is consistent with an increase in longshore drift (east to west) with sediment being
accumulated to the east of Wilson's Promontory. We should also note that this area east of Wilson
Promontory is one of the few estuarine environments along the entire Victorian coast and hence
some of the observed progradation may be due to fluvial deposits and ebb-tide delta formation
(Konlechner, et al., 2020).
The results above use the percentage of coastal locations prograding or receding as the measure of
whether the beach is responding to long term changes in waves and/or storm surge. As such, it
does not consider the magnitudes of the progradation or recession. Figure 6 shows histograms of
the magnitudes of the progradation/recession rates for each coastal sections. The figure confirms
the results above showing sections (c) 142E° – 144E° and (f) 148E° – 150E° are clearly receding
with other sections less clear, as explained for each section above.
In the above analysis, we speculate that changes in wave energy flux, $\Delta C_g E$ and mean wave
direction, $\Delta\theta$ are the primary drivers of the observed changes in shoreline. The observed data
supports this speculation. The Supplementary Material shows plots similar to Figures 3 and 4 for
changes in the other related quantities: significant wave height, $\Delta H_s$ (Figures S1 a-c and S1 d-f),
extreme significant wave height, $\Delta H_s^{95}$ (Figures S2 a-c and S2 d-f), mean wave period, $\Delta T_m$
(Figures S3 a-c and S3 d-f) and number of extreme wave events, $\Delta N_{H_s^{95}}$ (Figures S4 a-c and S4 d-
f).

**4. Discussion and conclusions**
Ghanavati, et al. (2023) found that at global scale, they could not distinguish a clear relationhip
between modelled (and observed) changes in wave energy flux and storm surge over the last 30


332 years and changes in shoreline postion. The present dataset extends this result by considering the
333 region of south-east Australia. This region is important in that it is an area with major spatial
334 variations in wave energy flux climate (mean conditions) and some of the largest coastal trends in
335 wave energy flux and mean wave direction globally in the last 30 years. In addition, both high
336 resolution coastal wave and storm surge hindcasts are available, as well as high resolution
337 observations of shoreline change. As such, this is a unique region to determine if observable
338 changes in shoreline position are evident as a consequence of long term changes in wave (and/or
339 storm surge) climate.

340 The results show clear changes in shoreline position, which are consistent with postive trends in
341 wave energy flux and changes in mean wave direction. In the western regions of the domain the
342 mean wave direction is from the south-west and there have been positive trends in wave energy
343 flux, $\Delta C_g E$ of approximately 14% (6/43kW/m). This appears to have resulted in non-stationary
344 longshore drift from west to east and shoreline changes of approximately 3m over the 30 year
345 study period.

346 In the central regions of the study domain both the mean wave energy flux and trends in wave
347 energy flux decrease, as the island of Tasmania provides protection from the south-westerly swell.
348 In this region there are no consistent trends in shoreline position with a similar number of coastal
349 locations receding and prograding. Although ocean beaches do show small recession.

350 To the eastern end of the study domain, the protection provided by Tasmania and the deepwater
351 conter-clockwise rotation of the mean wave climate means that the wave shadow of Tasmania
352 results in a relatively large counter-clockwise rotation of the mean wave direction (up to 6° over
353 the last 30 years). These changes in mean wave direction appear to be driving non-stationary
354 behaviour of the beach systems in the region with the coastline from 146° to 149° (approximately
355 300 km) receding by up to 30m over the 30 year study period.

356 The results presented in this analysis are consistent with a study of this same region by Konlechner,
357 et al. (2020) using lower resolution shoreline change data (Luijendijk, et al., 2018). The shoreline
358 change "hot-spots" of that study are consistent with the present results. The results of the present
359 study are also consistent with the global findings of Ghanavati, et al. (2023). Here, we find that
360 long term changes in wave climate can apparently drive long-term changes in beach location but
361 that relatively large changes in wave energy flux and/or direction are required to produce
362 measurable changes in beach position. As noted, the study region has both a very energetic wave
363 climate and some of the largest trends in this climate of any coastline.

364 Even in such a region, the observed changes in wave climate over the last 30 years are such that
365 the resulting changes in beach location are not large (up to 1.0 m/year or 30m over the study
366 period).

367 In the present analysis, we speculate that the observed changes in shoreline position in the western
368 section of the domain are driven by non-stationary longshore drift from west to east with sediment



transport being intercepted by Cape Bridgewater. Such behaviour is consistent with the observed
increases in wave energy flux and the predominately south-westerly swell. In the eastern sections
of the domain, we speculate that there is sediment transport from the east to west, intercepted by
Wilson's Promontory. This speculation is consistent with the predominately south-easterly swell
in the region and the observed couter-clockwise change in mean wave direction over the study
period.
Although such speculation is consistent with the datasets, other processes may also have an impact
on shoreline change. The most obvious such changes is sea level rise, which could be expected to
cause shoreline recession. Observations (Watson, et al., 2015; Nerem, et al., 2018) indicate that in
recent years sea level rise in the Australia region has been approximately 3mm/year. The bed slope
along the south-eastern coast of Australia is on avaerage approximately 1:100 (Athanasiou, et al.,
2019). Therefore, application of Bruun's rule (Bruun, 1962) would suggest a uniform recession of
approximately 0.3 m/year. Such a value is smaller than, but comparable, to the observed recession
in the western and eastern portions of the study domain. Recession due to sea level rise, however,
would not account for the observed progradation west of Cape Bridgewater or east of Wilson's
Promontory. In addition, Bishop-Taylor, et al. (2021) indicate that over their full dataset for
Australia, approximately the same number of beaches are receding (11.1%) as prograding (11.0%).
Table 1 indicates that for the present study region this is also the case. Sea level rise would be
expected to result in a net recession of beaches. In contrast non-equilibrium longshore drift driven
by changes in wave climate will cause some beaches to recede whilst other prograde.
Therefore, we conclude that the present results are more consist with the impacts of changes in
wave climate rather than sea level rise. Of course, sea level rise will undoubedly have a major
inpact in coming years.

**Code/Data availability**

All data used in the paper and codes for the analysis are available from the authors upon request.

**Competing Interests**

The authors declare no competing interests.

**Author Contributions**

MG: Data curation, Investigation, Writing – original draft, Writing – review and editing; IY:
Conceptualization, Investigation, Supervision, Writing – original draft, Writing – review and
editing; EK: Writing – review and editing; JL: Writing – review and editing





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

535    228.









**Tables and Figures**

| Coastal Segment | Recession (-0.05 to -1m/yr ) | Progradation (+0.05 to +1m/yr) | Stable (-0.05 to +0.05m/yr) |
|---|---|---|---|
| (a)  138°-140° | 40% | 45% | 15% |
| (b)  140°-142° | 40% | 46% | 14% |
| (c)  142°-144° | 53% | 27% | 20% |
| (d)  144°-146° | 37% | 49% | 14% |
| (e)  146°-148° | 40% | 50% | 10% |
| (f)  148°-150° | 60% | 30% | 10% |

Table 1: Percentage of coastal locations, as defined by the Bishop-Taylor, et al. (2021) dataset receding (-0.05 to -1.00m/year), prograding (+0.05 to +1.00m/year) or stable ( ± 0.05m/year) over the period 1988 to 2013.


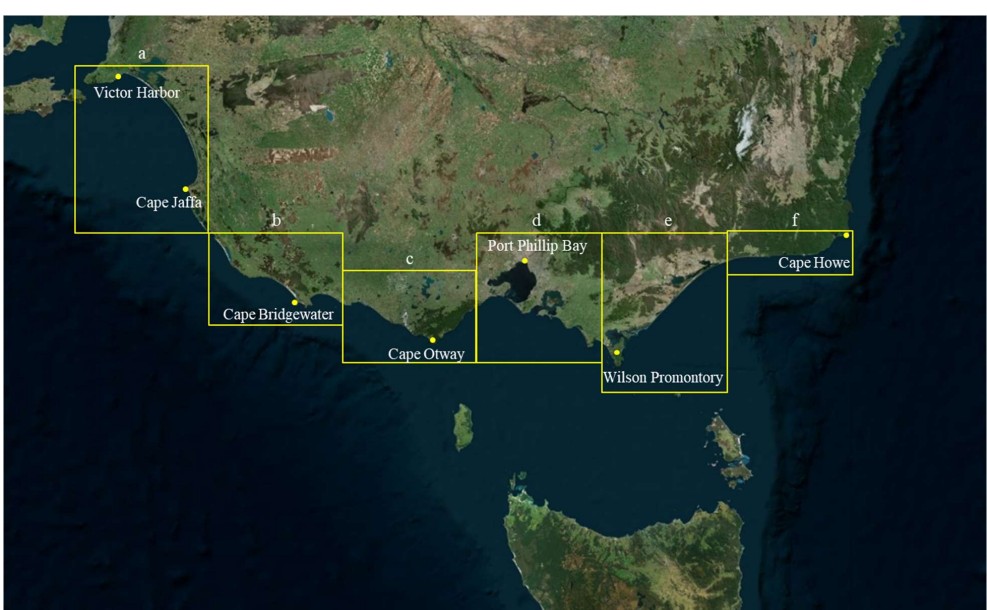

Figure 1: The coastal region of south-east Australia comprising the study area. For analysis
purposes the region is divided into six sections: (a) 138°-140°, (b) 140°-142°, (c) 142°-144°, (d)
144°-146°, (e) 146°-148° and (f) 148°-150° from west to east. The island of Tasmania is to the
south of this coastline. (© Google Maps)
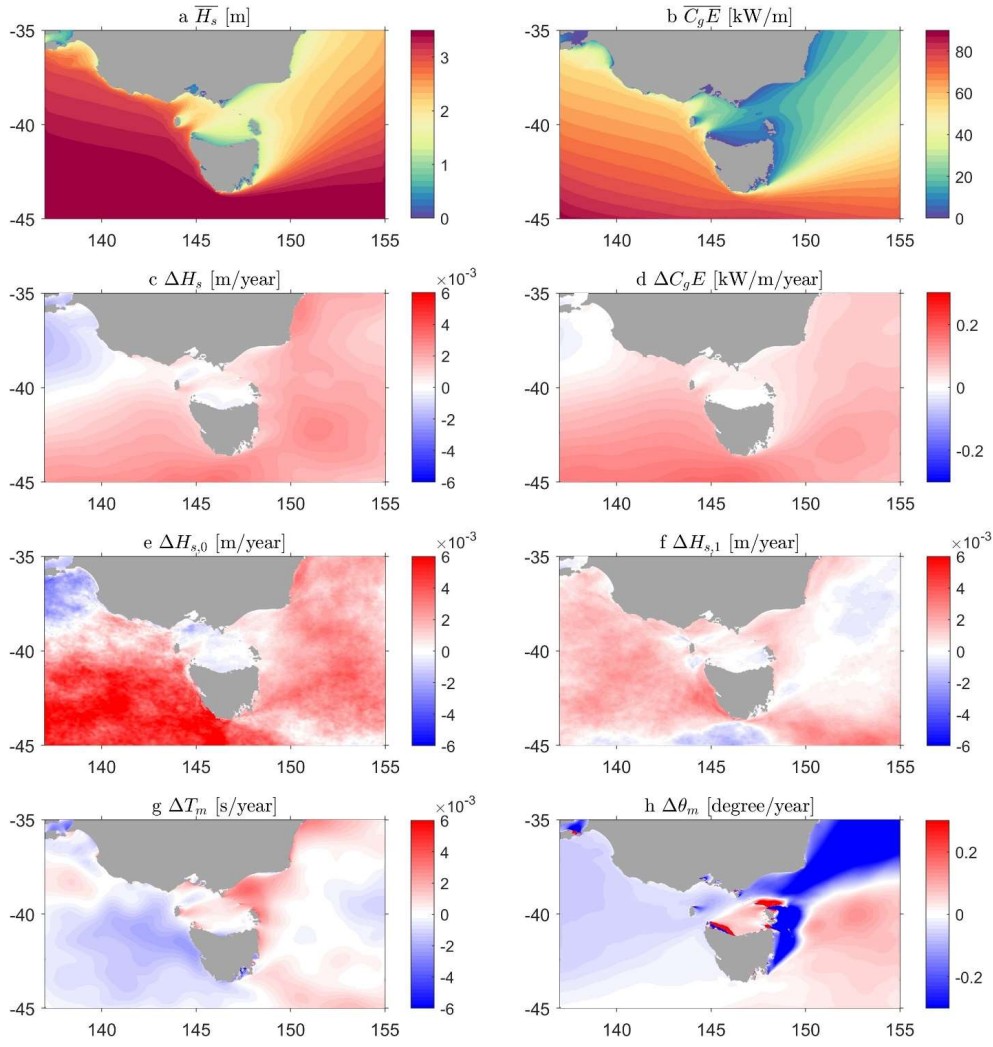


Figure 2: Wave climate and trends in the study region of south-eastern Australia over the period 1988 to 2013 as modelled by the Liu, et al. (2022) regional wave model. (a) mean significant wave height, (b) mean wave energy flux, (c) trend in significant wave height, (d) trend in wave energy flux, (e) trend in wind-wave portion of the spectrum, (f) trend in swell portion of the spectrum, (g) trend in mean wave period, (h) trend in mean wave direction.


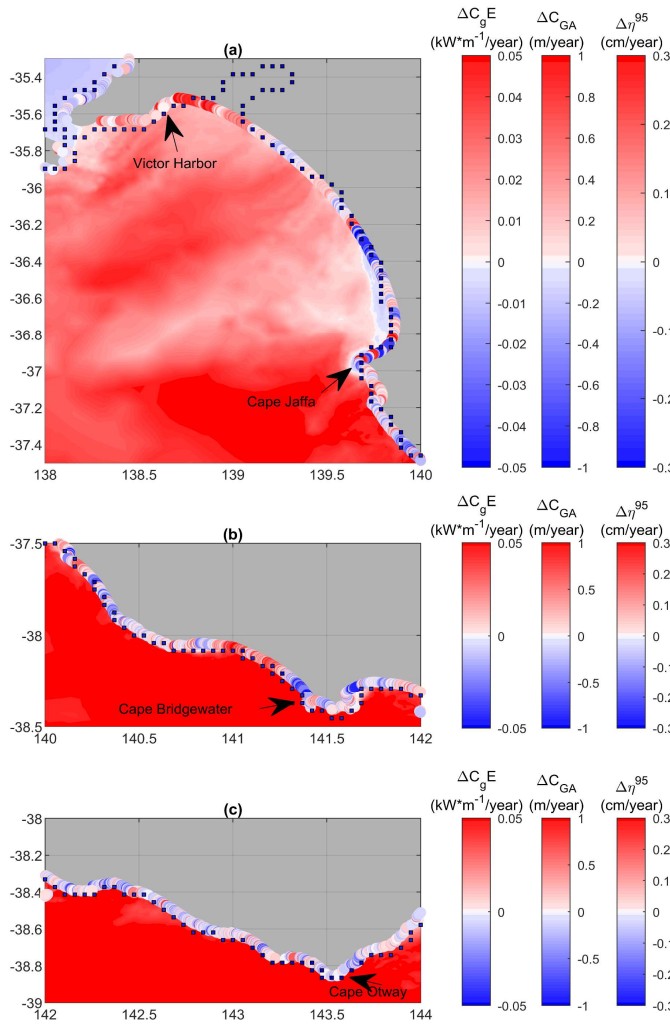

559

Figure 3 a-c: Trends in: wave energy flux, $\Delta C_g E$ shown as colour shaded values over the

domain, storm surge, $\Delta \eta^{95}$ shown as colour shaded squares at coastal model locations and

shoreline progradation/recession, $\Delta C_{GA}$ shown as colour shaded circles at beach locations.

Results shown for sections (a) 138E°-140E°, (b) 140E°-142E°, (c) 142E°-144E°.

564

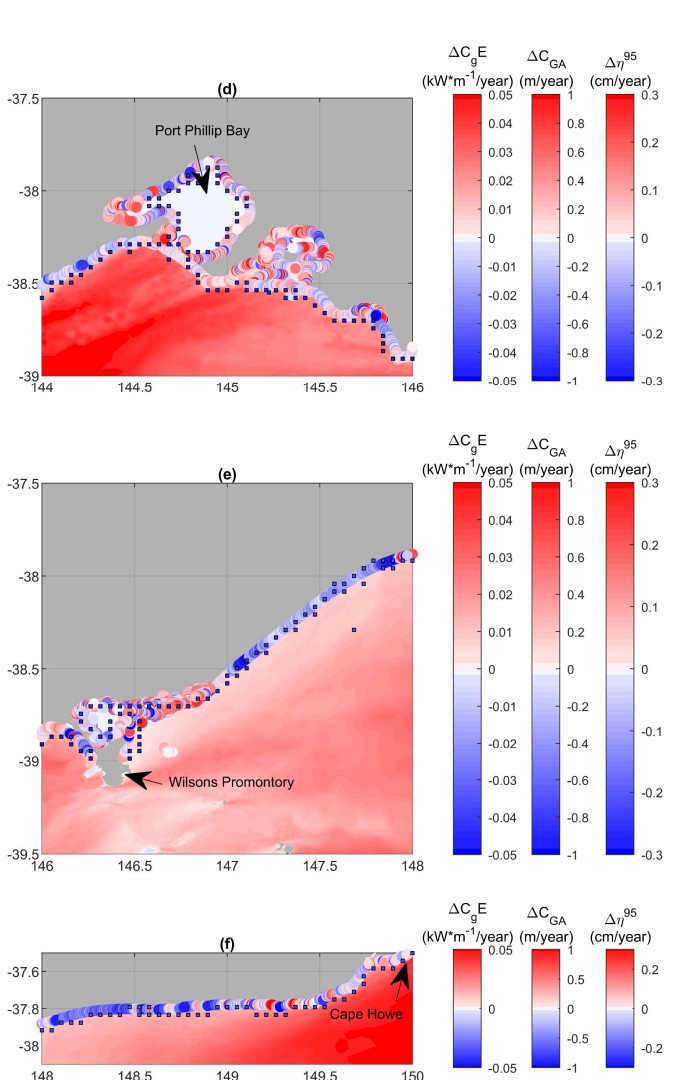

Figure 3 d-f: Trends in: wave energy flux, $\Delta C_g E$ shown as colour shaded values over the
domain, storm surge, $\Delta \eta^{95}$ shown as a colour shaded squares at coastal model locations and
shoreline progradation/recession, $\Delta C_{GA}$ shown as colour shaded circles at beach locations.
Results shown for sections (d) 144E°-146E°, (e) 146E°-148E° and (f) 148E°-150E°.


578

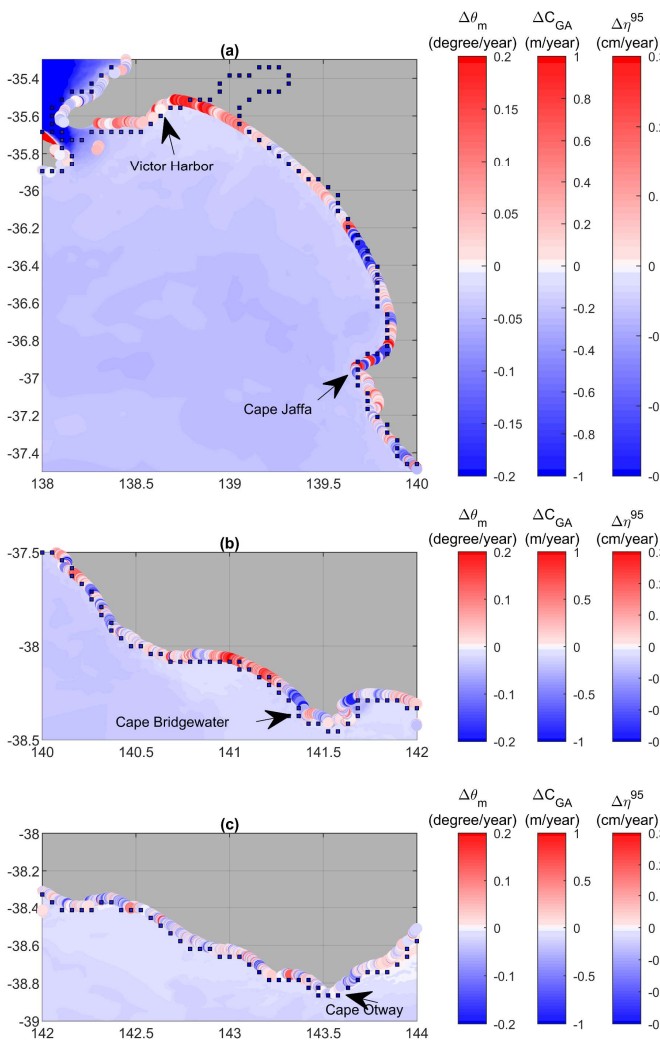

Figure 4 a-c: Trends in: mean wave direction, $\Delta\theta_m$ shown as colour shaded values over the
domain, storm surge, $\Delta\eta^{95}$ shown as colour shaded squares at coastal model locations and
shoreline progradation/recession, $\Delta C_{GA}$ shown as colour shaded circles at beach locations.
Results shown for sections (a) 138E°-140E°, (b) 140E°-142E°, (c) 142E°-144E°.

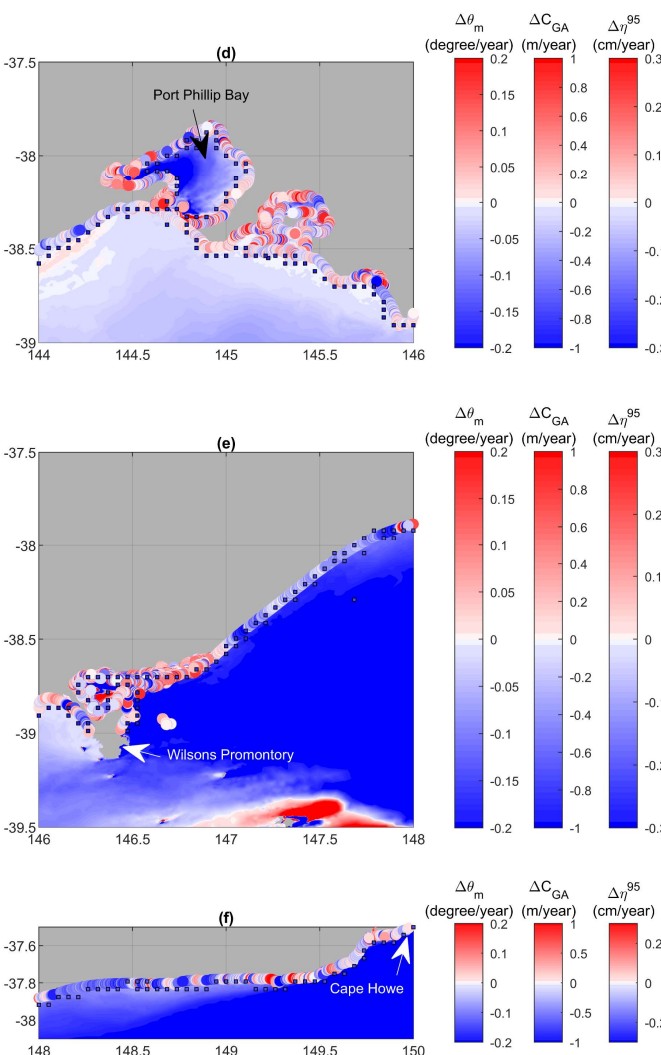

Figure 4 d-f: Trends in mean wave direction, $\Delta\theta_m$ shown as colour shaded values over the
domain, storm surge, $\Delta\eta^{95}$ shown as a colour shaded squares at coastal model locations and
shoreline progradation/recession, $\Delta C_{GA}$ shown as colour shaded circles at beach locations.
Results shown for sections (d) 144E°-146E°, (e) 146E°-148E° and (f) 148E°-150E°.



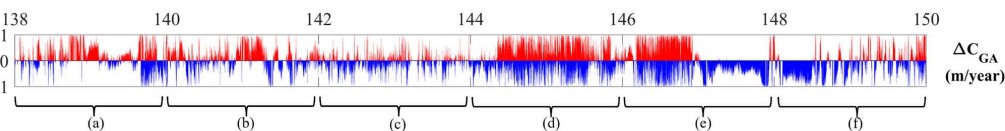

Figure 5: Bar chart showing values of progradation (red) and recession (blue), $\Delta C_{GA}$ at each coastal location of the Bishop-Taylor, et al. (2021) dataset. Values are shown as a function of the longitude (horizonal axis) and units are m/year. The regions shown in Figure 1 are labelled (a) to (f).


599

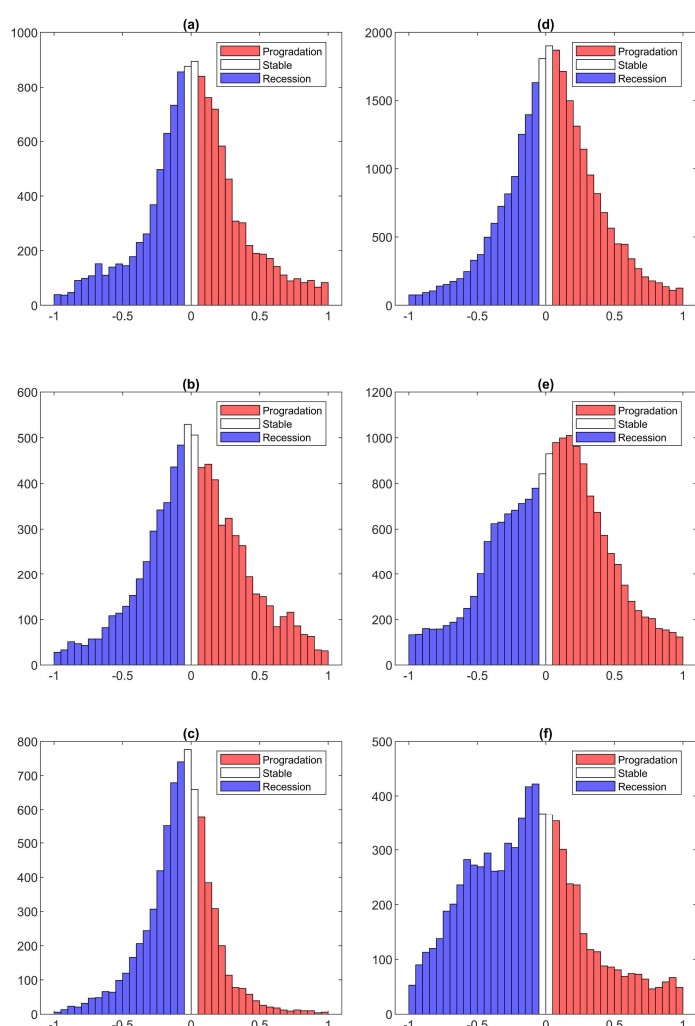

Figure 6: Histograms of progradation/recession rates for each of the coastal sections over the
period 1988 to 2013. (a) 138E°-140E°, (b) 140E°-142E°, (c) 142E°-144E°, (d) 144E°-146E°, (e)
146E°-148E° and (f) 148E°-150E° from west to east.