# Peer review of "The impact of long-term changes in ocean waves and storm surge on coastal shoreline"

_Natural Hazards and Earth System Sciences, 2023_

## Author Response (AR1)

**Editor's Comments Reply**

We would like to thank the Editor for the very supportive and constructive comments on our manuscript.

Below, we repeat each of the Editor's comments and respond. Our responses begin with the symbol "**" and are shown in *(red)* italics. All line numbers quoted in our reply refer to the revised version of the manuscript developed in response to the Reviewer and Editor's comments.

Thank you for the submission of your fascinating manuscript "The impact of long-term changes in ocean waves and storm surge on coastal shoreline change: A case study of Bass Strait and south-east Australia" (NHESS-2023-205).

As you know, two reviewers have now provided detailed reviews, to which you have replied in thoughtful detail. Both reviewers recommended minor revisions (although I have a feeling some of the revisions suggested are not so minor--they may take a bit of time!), and therefore I would like to invite you to submit a revised version of your manuscript.

Would you please also provide an 'author's reply' to the reviewers (feel free to use the same words that you used in what you have already uploaded). Please can you also include a track changes document between the old manuscript and the new one (you can include this as part of your 'author's reply').

*\*\*Thank you for these positive comments. We have completed the responses to each Reviewer and updated the manuscript accordingly.*

In addition to the suggestions from the reviewers, I would like to suggest the following 'major' issue:

(a) NOVELTY OF YOUR STUDY: Your research on 'ocean waves and storm surge on coastal shoreline change' is an interesting case study, but you do not tell us what is novel. This needs to be done both in the introduction so we understand and also in the discussion. You extended the global-level assessment to this specific case from your previous studies, which is inadequate to consider as a novel study. If you were to explain the results of your case study to someone in another country, what would they gain from your case study? Do they learn from your methodology and what you encountered when applying it? What is novel and what might they learn?

*\*\*Thank you for this important comment. In hindsight, we agree with the editor that we did not adequately position the importance of our study. Commencing in about 2005, the field of ocean wave climate has grown in popularity and now represents a relatively large field of research. Much of this research has been focussed on the role that long-term climate change plays in impacting the global wave climate. It has now been clearly shown that climate change over the last 30 years has resulted in stronger winds and hence larger waves. Projections for the future, show this situation*

*will continue in a warming world. These changes are largest in the Southern Hemisphere. One of the strong motivators of such work has been the inference that larger waves will give rise to enhanced impacts on sandy coastlines. If this is the case, then this may be an important element to consider in future coastal development, along with sea level rise. However, to date there have been no studies to investigate the potential relationship between long-term trends in wave climate and changes in beach systems.*

*We first tried to investigate the existence of such a relationship at global scale in Ghanavati et al. (2023). However, the study was inconclusive, most likely due to the lack of resolution in our datasets required at global scale. This was the motivator for the present study. We have moved to a regional study, not because we are particularly interested in this region, as such. Rather, we do this for two reasons: (a) This region has experienced some of the largest changes in wave height trends of any coastal region, globally . Thus, if there is a causal relationship it should be visible here and (b) At regional scale we can use high resolution wave, storm surge and beach position datasets. Our result shows, for the first time, a causal relationship. The increase in wave climate caused by a warming climate is impacting coastlines. Thus, the regional study has global importance and this study is both novel and important.*

***To address these issues, we have made the following changes to the manuscript.*

*** In response to the Reviewers and the Editor, the Abstract has been rewritten as.*

****" Numerous studies have demonstrated that significant global changes in wave and storm surge conditions have occurred over recent decades and are expected to continue out to at least 2100. This raises the question as to whether the observed and projected changes in waves and storm surges, will impact coastlines in the future? Previous global-scale analyses of these issues have been inconclusive. This study investigates the south-east coast of Australia over a period of 26 years (1988-2013). Over this period, this area has experienced some of the largest changes in wave climate of any coastal region, globally. The analysis uses high-resolution hindcast data of waves and storm surge, together with satellite observations of shoreline change. All datasets have been previously extensively validated against in situ measurements. The data are analysed to determine trends in each of these quantities over this period. The coastline is partitioned into regions and spatial consistency between trends in each of the quantities investigated. The results show that beaches along this region appear to have responded to the increases in wave energy flux and changes in wave direction. This has enhanced non-equilibrium longshore drift. Long sections of the coastline show small but measurable recession before sediment transported along the coast is intercepted by prominent headlands. The recession is largest where there are strong trends of increasing wave energy flux and/or changes in wave direction, with recession rates of up to 1m/year. Although a regional study, this finding has global implications for shoreline stability in a changing climate."*

***Lines 54-80 in the Introduction now state:*

*\*\*" If sandy coasts are impacted by changes in wave and storm surge conditions, the potential for continued increases in the values of these variables in the future raises the question as to what impact this may have on sandy coastlines and associated communities. As a means of determining potential future impacts, the obvious precursor is to assess the impacts that historical changes in long-term wave and storm surge conditions have had on coastlines. In the first study of its type, Ghanavati, et al. (2023) investigated this issue at global scale by using long-term modelled wave and storm surge data together with satellite observations of beach recession/progradation over the last 30 years. They found that, noting the relatively small trends in wave and storm surge conditions over this period, the accuracy of the available data, and other unrelated impacts on shoreline response (e.g. availability of sediment, human impacts), no clear relationship was evident.*

*In order to address the limitation of the Ghanavati, et al (2023) work, the present study examines, in much finer detail, the south-east coastline of Australia. This is an area where long-term trends in wave conditions are some of the largest in the world, responding to changes in wave climate in the Southern Ocean (Liu, et al., 2022). Therefore, if there is a causal link between changes in long-term wave and storm surge climate and shoreline response, one would expect clear signs in this region. As a regional area is considered, it is possible to use higher resolution data (both model and satellite) removing uncertainties in the global-scale Ghanavati, et al (2023) study. In addition, the regional-scale study enables an analysis of the role beach compartments play in defining sediment transport. As such, one can investigate changes in longshore drift due to changes in wave climate and the characteristic signature of such non-equilibrium transport with eroding beaches and deposition of sediment behind peninsulas.*

*Although the present study is regional, the area being studied is a proxy for the potential impacts one may see in other regions of the world as changes in wave and storm surge climate are projected to continue to change in the future. Hence, the finding of the study have global implications for shoreline response in the future. The study is unique in that it has been possible to combine high resolution datasets for waves, storm surge and shoreline response and addresses a previously unexplored area of shoreline response in a changing climate."*

*\*\*Lines 411-417 in the Conclusions now state:*

*\*\*"Although the present study is regional, it has global implications for the magnitude of changes in shoreline response which may result in other regions of the world under future projections of changes in wave climate. The present study clearly shows that impacts of changing wave climate will have strong regional characteristics and that it is important to consider the unique nature of each region in determining potential impacts. The response to individual coastal compartments will differ in terms of the magnitude of the response and even the sign (recession verses progradation)."*

(b) BROADER CONTEXT OF YOUR STUDY. You do not relate your work to the broader literature of what others have done. We need to understand this broader

context and what others have done.

*\*\*Thank you for this comment. The Introduction from Lines 27-80 contains an extensive literature review which we believe now comprehensively positions the present study within previous work. This review considers the field in a systematic manner.*

- *It looks at the spatial and temporal scales that impact coastlines, considering work that has investigated changes on the time scales of days and multiple years (climate oscillations).*
- *It looks at trends observed in waves, storm surge and sea level*
- *Finally, it looks at studies dealing with shoreline changes as a result of long-term climate changes in waves and storm surges. To the best of our knowledge, there is only one – our previous inconclusive study Ghanavati et al. (2023).*

*\*\*There are a total of 45 citations in this literature review. We believe this new introduction comprehensively positions the work within previous studies and shows the unique nature of the present study.*

**Reviewer 1 Reply**

We would like to thank Reviewer 1 for the very positive assessment of the manuscript and the helpful suggestions for improvement. Below, we repeat each of the reviewer comments and respond. Our responses begin with the symbol "\*\*" and are shown in *(red)* italics. All line numbers quoted in our reply refer to the revised version of the manuscript developed in response to the reviewer comments.

The manuscript investigates the possible causes of long-term changes in the coastline in Australia, analysing data of wave climate, storm surges and coastline position over the past few decades. The data oceanographic data mostly stem from simulations with ocean models with data assimilation or forced by observed atmospheric forcing.

The main conclusion is that, in this part of the world, the changes in the coastlines can be attributed to changes in the wave intensity and direction and possibly also to changes of storm surges, whereas the impact of sea level rise is more questionable, as it does not agree with a general retreat of the coastline over all areas analysed.

Recommendation: I had the impression that this manuscript version has already undergone a previous peer review. However, I see it for the first time, so I am unaware of previous exchanges between the authors and reviewers.

In general, the manuscript is already in good shape. It is well-written, and the structure is adequate. The analysis is also technically sound. The conclusions are also logically derived from the data analysis.  I have some minor comments that the authors may want to consider, in particular concerning the clarity of the abstract. My recommendation is that the manuscript requires only minor changes.

 **We thank Reviewer 1 for the positive assessment of the quality of the manuscript.*

Main point

1) The abstract devotes too much space for general comments on climate changes and changes of the wave climate, which actually would fit better in the introduction, but it is rather uninformative in terms of methods, data sets used and results. For instance, the abstract does not mention which type of data has been used and which methods have been applied to derive the conclusions. The study's conclusions are very briefly mentioned - more details would make the abstract much more informative. It does not mention sea-level rise to confirm or rule out it  as a driving force for coastline change.

**In hindsight, we agree with the reviewer and the abstract has been re-written as follows.

*Numerous studies have demonstrated that significant global changes in wave and storm surge conditions have occurred over recent decades and are expected to continue out to at least 2100. This raises the question as to whether the observed and projected changes in waves and storm surges, will impact coastlines in the future? Previous global-scale analyses of these issues have been inconclusive. This study investigates the south-east coast of Australia over a period of 26 years (1988-2013). Over this period, this area has experienced some of the largest changes in wave climate of any coastal region, globally. The analysis uses high-resolution hindcast data of waves and storm surge, together with satellite observations of shoreline change. All datasets have been previously extensively validated against in situ measurements. The data are analysed to determine trends in each of these quantities over this period. The coastline is partitioned into regions and spatial consistency between trends in each of the quantities investigated. The results show that beaches along this region appear to have responded to the increases in wave energy flux and changes in wave direction. This has enhanced non-equilibrium longshore drift. Long sections of the coastline show small but measurable recession before sediment transported along the coast is intercepted by prominent headlands. The recession is largest where there are strong trends of increasing wave energy flux and/or changes in wave direction, with recession rates of up to 1m/year. Although a regional study, this finding has global implications for shoreline stability in a changing climate.*

There are also some minor grammatical errors in the abstract, with commas misplaced, as indicated below.

*\*\*Corrected in revised abstract.*

Minor points

2) 'storm surges, the question of whether the observed and projected changes in waves and storm surges, will impact coastlines in the future, is important.'

Delete comma after 'surges'

*\*\*Text removed in new version of abstract.*

3) 'At time scales of 2 to 10 years, changes in storminess associated with climate indices (e.g. El Niño) '

I guess that storminess impacts the coastline independently of whether it is associated with El Niño or not. The sentence is a bit confusingly formulated.

*\*\*The point we were trying to make is that, for instance, during an El Niño there may be enhanced storminess, which would result in a sustained impact on the coastline. This is in contrast to more normal periods where storms may impact a coastline but the beaches will then recover in calmer periods. To clarify the point, Lines 33-36 have been re-written as:*

*\*\*At time scales of  2 to 10 years, changes in storminess associated with climate indices (e.g. El Niño) (Ranasinghe, et al., 2004; Harley, et al., 2011; Barnard, et al., 2015; Vos, et al., 2023) can result in sustained impacts on beach systems.*

4) 'the potential for continued increases in the values of these variables in the future, raises the question as to what impact this may have on sandy coastlines and associated communities'

Delete comma after 'future'

*\*\*Deleted, as suggested, L55.*

5) Regional Ocean Modelling System.

Was observed sea level prescribed as a boundary condition, and if yes, in which way?

*\*\*To clarify this point, the following text has been added at L134-137.*

*\*\*Tidal currents and heights at open boundaries were specified from the TPXO7.2 global model (Egbert & Erofeeva, 2002). TPXO7.2 best fits (in a least squares sense) the Laplace tidal equations and along track averaged data from TOPEX/Poseidon and Jason altimetry data.*

6) Bishop-Taylor et al. data

The paragraph indicates the along-shore spatial resolution, but the accuracy of the data (uncertainty range) is also very important to establish changes in the coastline

*\*\*The following text has been added at L147-148 to clarify this point.*

*\*\* The Mean Absolute Error (MAE) in the trend across these validation points was 0.35 m/year (Bishop-Taylor, et al., 2021).*

7) 'Of course, sea level rise will undoubedly have a major impact in coming years.'

undoubtedly, impact

This assertion might be true, but it appears to be speculative in this study. Actually, the opposite conclusion seems more logical, given that storminess and changes in wave climate have been more important in the past. In which way can sea level rise become more important than the other factors in the future, and why?

*\*\*This is a good point and to avoid unwarranted speculation, the sentence: "Of course, sea level rise will undoubtedly have a major impact in coming years." Has been deleted at L410.*

**Reviewer 2 Reply**

We would like to thank Reviewer 2 for the very positive assessment of the manuscript and the helpful suggestions for improvement. Below, we repeat each of the reviewer comments and respond. Our responses begin with the symbol "\*\*" and

are shown in *(red)* italics. All line numbers quoted in our reply refer to the revised version of the manuscript developed in response to the reviewer comments.

Review on "The impact of long-term changes in ocean waves and storm surge on coastal shoreline change: A case study of Bass Strait and south-east Australia" by Mandana Ghanavati, Ian R. Young, Ebru Kirezci, and Jin Liu.

The manuscript offers an extensive examination of observed changes in shoreline position within the context of long-term variations in wave climate in south-east Australia. The primary finding is an association between long-term changes in wave climate, specifically in wave energy flux and mean wave direction, and tangible alterations in shoreline position. While the manuscript is generally well-constructed, I have a few comments that could enhance its clarity and depth.

*\*\*Thank you for the supportive comments*

**Abstract:** The abstract, though generally clear, requires clarification in certain terms such as "significant global changes," "such changes," and "these issues." Providing more explicit definitions or rephrasing these terms would improve comprehension.

*\*\*As noted in the reply to Reviewer 1, the abstract has been re-written to address these issues.*

*\*\*Numerous studies have demonstrated that significant global changes in wave and storm surge conditions have occurred over recent decades and are expected to continue out to at least 2100. This raises the question as to whether the observed and projected changes in waves and storm surges, will impact coastlines in the future? Previous global-scale analyses of these issues have been inconclusive. This study investigates the south-east coast of Australia over a period of 26 years (1988-2013). Over this period, this area has experienced some of the largest changes in wave climate of any coastal region, globally. The analysis uses high-resolution hindcast data of waves and storm surge, together with satellite observations of shoreline change. All datasets have been previously extensively validated against in situ measurements. The data are analysed to determine trends in each of these quantities over this period. The coastline is partitioned into regions and spatial consistency between trends in each of the quantities investigated. The results show that beaches along this region appear to have responded to the increases in wave energy flux and changes in wave direction. This has enhanced non-equilibrium longshore drift. Long sections of the coastline show small but measurable recession before sediment transported along the coast is intercepted by prominent headlands. The*

*recession is largest where there are strong trends of increasing wave energy flux and/or changes in wave direction, with recession rates of up to 1m/year. Although a regional study, this finding has global implications for shoreline stability in a changing climate.*

**Introduction:** It would be better to have a more explicit statement on why previous global-scale analyses were inconclusive and how this study addresses those limitations. This addition would better contextualize the significance of the current research.

*\*\*This is an important point and the Introduction has been changed to reflect these points. L59-74 are reproduced below, with L70-74 added to clarify the point made by Reviewer 2.*

*\*\* "Ghanavati, et al. (2023) have investigated this issue at global scale by using long-term modelled wave and storm surge data together with satellite observations of beach recession/progradation over the last 30 years. They found that, noting the relatively small trends in wave and storm surge conditions over this period, the accuracy of the available data, and other related impacts on shoreline response (e.g. availability of sediment, human impacts), no clear relationship was evident.*

*The present study extends the Ghanavati, et al (2023) work by examining the south-east coastline of Australia in detail. This is an area where long-term trends in wave conditions are some of the largest in the world, responding to changes in wave climate in the Southern Ocean (Liu, et al., 2022). As a regional area is considered, it is possible to use higher resolution data (both model and satellite) removing uncertainties in the global-scale Ghanavati, et al (2023) study. In addition, the regional-scale study enables an analysis of the role beach compartments play in defining sediment transport. As such, one can investigate changes in longshore drift due to changes in wave climate and the characteristic signature of such non-equilibrium transport with eroding beaches and deposition of sediment behind peninsulas."*

**Methodology:** It would be valuable to understand the approach taken to calculate the shoreline response to waves and storm surges. In particular, the authors mentioned that the shoreline response (L56) has relatively small impact on the global scale. How do the authors justify the same statement for a local scale?

*\*\*As noted in Section 2.3, trend analysis of waves, storm surge and shoreline position are undertaken. The analysis then seeks to determine if there is a qualitative relationship between these quantities. One cannot undertake a simple correlation analysis, as longshore drift can result in some parts of the coast receding, whilst the sediment is deposited in other areas, which will prograde. Therefore, we examine the coastline in segments where the role of individual coastal compartments can be described (Section*

*3.3). The expanded Introduction (L59-74) now better describes this approach and why it differs from the global analysis.*

**Sentence Clarity:** Consider rephrasing the sentence at L364-366.

*\*\*The sentence has been added to the end of the preceding paragraph, to make it clear that it refers to this text and rephrased as (L384-386):*

*\*\* However, even in a region such as this, where long-term changes in wave energy flux are relatively large, the resulting changes in beach location are only approximately 1.0 m/year or 30m over the study period.*

**Future Research and Limitations:** Add a discussion for future research. Providing further clarification on spatial/temporal limitations and uncertainties would enhance the completeness of the work.

*\*\*The following section has been added at L418-437 to discuss possible future work.*

*\*\* As noted, the present analysis provides the first evidence of a causal relationship between long-term climate trends in waves and shoreline change. It does, however, have a number of limitations which should be addressed in future research if a comprehensive understanding of the impacts future projected changes in wave climate may have on our coastlines. These future studies could include:*

*•      Detailed sediment transport modelling to assess whether the observed changes in wave energy flux and wave direction would be expected to result in non-stationary longshore drift of the magnitude observed in the recorded shoreline position.*

*•      The extraction of shoreline position from relatively low-resolution satellite images is computationally challenging. The Bishop-Taylor, et al. (2021) dataset represents a significant advance in resolution and accuracy. Further developments in the use of Artificial Intelligence approaches to determining shoreline position are expected to further reduce errors in such data.*

*•      The present analysis is limited to south-east Australia as there were opportunistic high-resolution datasets of long-term changes in waves, storm-surge and shoreline position available. Dedicated projects modelling specific areas for the purpose of better determining the relationships between changes in these quantities would better quantify the likely impacts of future changes on vulnerable shoreline.*

**References:** References at L460-468 and L475-480. Some items were not mentioned in the manuscript, and there is confusion regarding citations.

*\*\*These references are all mentioned in the text:*

*Liu, J. et al. (2022) – L47*

*Liu, J. et al. (2023) – L53*

*Liu, Q. et al. (2021) – L118*

*Meucci et al. (2023) – L51 and L52*

---

## Referee Report (RR1)

I would like to express my appreciation to the authors for their efforts in revising the manuscript. I have several comments and suggestions for improvement, including the correction of typos and clarity issues.

Comments:
- Line 46: Remove the brackets "2021)(Young".
- Line 64: Replace "adress" with "address".
- Line 103: Replace "reanalyzes" with "reanalyses".
- Lines 351-354: There are typos such as "relationhip" and "postion". Additionally, the sentence "The present dataset extends this result" seems problematic, as "this result" from the previous research is inconclusive. It would be more logical to avoid "extends this result".
- Line 358: Change "observations of shoreline change" to "observation of shoreline changes".
- Line 393: Replace "couter-clockwise" with "counter-clockwise".
- Line 399: Replace "avaerage" with "average".
- Line 419: The sentence "It does, however, have a number of limitations which should be addressed in future research if a comprehensive understanding of the impacts future projected changes in wave climate may have on our coastlines" is unclear. I recommend rephrasing it for better clarity.
- Line 429: Replace "resoltion" with "resolution".
- Line 433: The term "opportunistic" is unclear in its context. It would be beneficial to provide clarification or rephrase the sentence for better understanding.

---

## Author Response (AR2)

We would like to thank the Reviewers for their supportive comments and their diligent reading of the revised paper. We appreciate the editorial comments and suggestion which we have incorporated into the updated draft of the manuscript.

**Reviewer 1**

*The y-axis in Figure 5 is too short, and there is no obvious reason why it should be that short. Could it be doubled? I think that would help the reader.*

This is a good point and the axis has been doubled in size, as suggested.

**Reviewer 2**

*I would like to express my appreciation to the authors for their efforts in revising the manuscript.*

*I have several comments and suggestions for improvement, including the correction of typos and clarity issues.*

*Comments:*

*● Line 46: Remove the brackets "2021)(Young".*

*● Line 64: Replace "adress" with "address".*

*● Line 103: Replace "reanalyzes" with "reanalyses".*

Above points corrected.

*● Lines 351-354: There are typos such as "relationhip" and "postion". Additionally, the sentence "The present dataset extends this result" seems problematic, as "this result" from the previous research is inconclusive. It would be more logical to avoid "extends this result".*

Corrected and updates, as suggested.

*● Line 358: Change "observations of shoreline change" to "observation of shoreline changes".*

*● Line 393: Replace "couter-clockwise" with "counter-clockwise".*

*● Line 399: Replace "avaerage" with "average".*

Above points corrected.

*● Line 419: The sentence "It does, however, have a number of limitations which should be addressed in future research if a comprehensive understanding of the*

*impacts future projected changes in wave climate may have on our coastlines" is unclear. I recommend rephrasing it for better clarity.*

The words "if a comprehensive understanding of the impacts future projected changes in wave climate may have on our coastlines" have been deleted.

*● Line 429: Replace "resoltion" with "resolution".*

Corrected.

*● Line 433: The term "opportunistic" is unclear in its context. It would be beneficial to provide clarification or rephrase the sentence for better understanding.*

Sentence rephrased as: "The present analysis is limited to south-east Australia, as there were high-resolution datasets of long-term changes in waves, storm-surge and shoreline position available for this region."